# *ENO1* Regulates Apoptosis Induced by Acute Cold Stress in Bovine Mammary Epithelial Cells

**DOI:** 10.3390/ani15172559

**Published:** 2025-08-31

**Authors:** Na Shen, Jie Wang, Jiayu Liao, Hengwei Yu, Wenqiang Sun, Xianbo Jia, Songjia Lai

**Affiliations:** College of Animal Science and Technology, Sichuan Agricultural University, Chengdu 611130, China; 13289700731@163.com (N.S.); m18483220592@163.com (J.W.); ljiayu0121@163.com (J.L.); 18792427097@163.com (H.Y.); wqsun2021@163.com (W.S.); jaxb369@sicau.edu.cn (X.J.)

**Keywords:** acute cold stress, bovine mammary epithelial cells (BMECs), apoptosis, enolase 1 (ENO1)

## Abstract

Acute cold stress adversely affects *dairy cow* health and lactation, but the specific mechanism by which damages bovine mammary epithelial cells (BMECs) remains unclear. Our study shows that acute cold stress harms BMECs by increasing cell apoptosis and disrupting the balance of apoptosis-related genes. We further identified that alpha-enolase (ENO1), a key glycolytic enzyme with multiple non-glycolytic functions in pathophysiological processes including signal transduction and apoptosis, plays a critical protective role, since its downregulation exacerbates apoptosis. These findings explain how acute cold stress impairs the normal function of BMECs and suggest a new direction for protecting *dairy cows* from cold stress by targeting *ENO1* as a potential target. This research could lead to better strategies for maintaining cow mammary health, ensuring stable milk production, and reducing economic losses for farmers in cold environments.

## 1. Introduction

Against the backdrop of intensifying global climate change and frequent extreme cold events, the adverse effects of cold environments on the cattle industry have become increasingly prominent. In major dairy-producing regions such as Russia and Ontario, Canada, extreme winter cold has emerged as a critical constraint on dairy industry development. Studies indicate that winter cold waves can induce cold stress responses in *dairy cows* [1]. This response exerts multifaceted detrimental effects on production performance. For one thing, it causes significant declines in milk yield; research by Thompson et al. has confirmed that reduced mammary blood flow under cold conditions may be a key mechanism inhibiting milk secretion [2]. For another, it increases the incidence of diseases such as mastitis, elevating morbidity and mortality rates in calves, and ultimately reducing economic returns [3,4].

Cold stress refers to a series of defensive responses and functional adjustments initiated by the body when animals are exposed to temperatures below their temperature tolerance threshold [5,6]. As an important type of environmental stress parallel to heat stress, both can activate the neuro-endocrine-immune network, trigger oxidative stress and metabolic disorders, and affect the physiological health of animals [7,8,9]. However, there are significant differences in physiological mechanisms between cold stress and heat stress. Heat stress is centered on heat dissipation disorders, where cell damage stems from high-temperature-induced protein denaturation, and cell homeostasis is mainly maintained by activating heat shock proteins (HSPs) [10,11]. In contrast, cold stress focuses on the surge in heat production demand, and copes with low temperatures by enhancing metabolic heat production and activating cold resistance-related genes [12,13,14].

Under normal circumstances, animals maintain stable core body temperature by enhancing metabolic heat production. However, when the ambient temperature is excessively low, it leads to an imbalance between heat production and heat dissipation, disrupting the homeostasis of the body’s internal environment and subsequently triggering physiological dysfunction [15]. Studies have revealed that cold stimulation directly reduces blood flow in the mammary tissues of ruminants. On one hand, this is because low temperatures activate the sympathoadrenal medullary axis to release norepinephrine, and simultaneously activate oxidative stress-related pathways, which further exacerbate norepinephrine-mediated vasoconstriction. On the other hand, to maintain core body temperature, the body prioritizes blood distribution to core organs, thereby reducing mammary blood flow [16,17,18,19]. Reduced mammary blood flow results in insufficient nutrient supply and impaired excretion of metabolic waste, placing *bovine* mammary epithelial cells (BMECs) in an adverse metabolic environment. This leads to mitochondrial dysfunction, insufficient ATP production, and triggers the mitochondrial apoptotic pathway. Additionally, direct damage caused by low temperatures further exacerbates BMEC apoptosis [20,21,22]. The phenomenon of cold stress-induced cell apoptosis has been confirmed in various cell types, including mouse lymphocytes and bovine Sertoli cells [23,24,25,26]. As a genetically regulated programmed death process, cell apoptosis is of great significance in maintaining tissue homeostasis and responding to environmental stress [27,28]. However, excessive apoptosis leading to large-scale cell death can have adverse effects on the organism [29,30].

To counteract cold stress-induced damage, cells activate intrinsic cold resistance mechanisms. Alpha-enolase (ENO1), a key enzyme in the glycolytic pathway, also possesses diverse non-glycolytic functions [31,32]. It is involved in various pathophysiological processes such as cellular signal transduction and apoptosis [33,34]. Existing studies have suggested that under environmental stress conditions, the expression level and functional status of *ENO1* may undergo adaptive changes, thereby affecting the process of cell apoptosis. For *dairy cows*, the number of mammary epithelial cells and their lactation capacity directly determine milk yield, and reducing their apoptosis rate is an important prerequisite for maintaining lactation performance [35].

However, research on the mechanisms underlying cold stress-induced apoptosis in bovine mammary epithelial cells (BMECs) remains limited, and the role of *ENO1* in this process is yet to be defined. Therefore, this study employed BMECs to establish an in vitro model simulating acute cold stress. Integrating transcriptomic analysis and functional validation, we systematically investigated the signaling pathways of acute cold stress-induced apoptosis and the core regulatory function of *ENO1*. Our findings will provide a theoretical basis for deciphering the apoptotic mechanisms in bovine mammary cells under acute cold stress, while also offering foundational insights for breeding cold-resistant dairy cattle lines and informing the development of cold-wave prevention strategies for dairy farms.

## 2. Materials and Methods

### 2.1. Cell Culture and Treatment

The bovine mammary epithelial cell line (BMECs, MAC-T) was cultured in DMEM/F12 medium supplemented with 10% fetal bovine serum (Gibco, Carlsbad, CA, USA) and 1% penicillin-streptomycin (Corning, Corning, NY, USA) at 37 °C under 5% CO_2_. To investigate the effects of acute cold stress on BMECs, cells at 85–90% confluence were seeded into 6-well or 12-well plates (Corning, Corning, NY, USA) and incubated overnight at 37 °C according to established protocols. The following day, cells were transferred to a 4 °C environment and subjected to cold stress treatment for 0, 2, 4, 6, or 8 h in medium containing 0.1 M HEPES (pH 7.4; Thermo Fisher Scientific, Waltham, MA, USA). Cells were harvested post-treatment for subsequent experiments. Based on combined expression analysis of HSP90 and apoptosis-related genes, the optimal condition for establishing the acute cold stress model was determined to be 4 h of exposure at 4 °C.

### 2.2. Preparation of Sequencing Samples

RNA sequencing (RNA-seq) analysis was performed on bovine mammary epithelial cells. This included three biological control replicates (CON1, CON2, CON3; cultured at 37 °C for 4 h) and three biological replicates of the cold stress-treated group (LT1, LT2, LT3; treated at 4 °C for 4 h). All sequencing services were provided by LC-Bio Technology Co., Ltd. (Hangzhou, China).

### 2.3. Apoptosis Assay

Cell apoptosis rates were determined using an Annexin V-FITC Apoptosis Detection Kit (Beyotime, Nantong, China). Following the manufacturer’s protocol, cells were collected after cold stress treatment, stained with Annexin V-FITC and propidium iodide (PI), and incubated at room temperature for 15 min. DAPI (Beyotime, Nantong, China) was added to stain nuclei. Cells were observed under an inverted fluorescence microscope (Olympus, Tokyo, Japan): nuclei appeared blue (DAPI), early apoptotic cells fluoresced green (Annexin V-FITC positive only), and late apoptotic cells fluoresced red (double positive for Annexin V-FITC and PI). At least three random fields per group were captured. Quantification using ImageJ2 software (National Institutes of Health, Bethesda, MD, USA) involved: (1) converting images to 8-bit grayscale; (2) applying a uniform threshold to all images to distinguish signal from background; (3) counting positive cells using the “Analyze Particles” tool, with size and circularity parameters set to exclude debris (0.5–50 μm^2^, circularity 0.3–1.0). Results are expressed as the percentage of PI- and FITC-positive cells relative to the total number of DAPI-stained nuclei.

### 2.4. Total RNA Extraction and RT-qPCR

Total RNA was extracted using TRIzol reagent, and its concentration and purity were measured. RNA samples were deemed qualified when the A260/A230 and A260/A280 ratios both exceeded 1.80. RNA integrity was assessed via 1.5% agarose gel electrophoresis; qualified samples exhibited clear, bright 28S and 18S rRNA bands, with the 28S band intensity approximately twice that of the 18S band. Qualified RNA was stored at −80 °C. Reverse transcription was performed using an mRNA Reverse Transcription Kit (Vazyme, Nanjing, China) according to the manufacturer’s instructions. Quantitative real-time PCR (RT-qPCR) was conducted using SYBR^®^ Premix Ex Taq™ (Foregene, Chengdu, China) on a ForeQuant F4 Sequence Detection System. The thermal cycling protocol was: 95 °C for 10 min (initial denaturation); 40 cycles of 95 °C for 10 s (denaturation) and 60 °C for 20 s (annealing/extension); followed by melt curve analysis to verify primer specificity. All primer sequences used in this study are listed in Table 1. The relative expression level of each gene was calculated using the 2^(−ΔΔCt)^ method, with β-actin as the internal reference gene.

### 2.5. siRNA Synthesis and Transfection

Small interfering RNAs (siRNAs) and negative control (NC) siRNA were designed and synthesized by GenePharma (Shanghai, China). Their sequences are listed in Table 2. For transfection, cells were seeded into 6-well plates. Upon reaching 50–60% confluence, siRNA (final concentration: 50 nM) and Lipofectamine 3000 (Invitrogen, Carlsbad, CA, USA) were mixed in Opti-MEM medium (Gibco, Grand Island, NY, USA) according to the manufacturer’s protocol. The mixture was incubated at room temperature for 15 min and then added to the cells. Cells were subsequently incubated at 37 °C for 48 h. Subsequent experiments were performed following verification of transfection efficiency by RT-qPCR or Western blotting.

### 2.6. Protein Extraction and Western Blotting

Cells were lysed using radioimmunoprecipitation assay (RIPA) buffer (Beyotime, Nantong, China). Protein concentration was determined with a BCA protein assay kit (Beyotime, Nantong, China). Protein expression levels were detected by Western blotting, with β-actin serving as the internal reference for normalization. Protein samples were separated using SDS-polyacrylamide gel electrophoresis (SDS-PAGE). Subsequently, separated proteins were transferred onto polyvinylidene difluoride (PVDF) membranes (Beyotime, Nantong, China), followed by a 2 h blocking step. After blocking, membranes were incubated with appropriate primary antibodies at 4 °C overnight (antibody dilution concentrations are listed in Table 3). Following secondary antibody incubation, protein bands were visualized using Sparkjade ECLsuper (ED0015-A; Sparkjade Biotechnology Co., Jinan, China).

### 2.7. Statistical Analysis

All data are presented as mean ± standard error of the mean (SEM). Experiments were independently repeated at least three times. Comparisons between two groups were analyzed using Student’s *t*-test, while one-way ANOVA was employed for multiple comparisons. Statistical analyses were performed using GraphPad Prism 9. Significance levels are denoted as follows: * *p* < 0.05, ** *p* < 0.01.

## 3. Results

### 3.1. Establishment of Acute Cold Stress Model and Apoptosis Assessment

After acute cold stress treatment, apoptosis rates, cold adaptation responses, and expression levels of apoptosis-related genes in BMECs were analyzed, with the following results: Annexin V-FITC assays revealed a highly significant increase in apoptosis rates at 4, 6, and 8 h post-cold stress compared to controls (*p* < 0.01, Figure 1A,B), with apoptosis severity escalating over time. This indicates a positive correlation between cold stress duration and apoptotic progression. RT-qPCR analysis of relevant genes (Figure 1C–G) revealed: Heat shock protein 90 (HSP90, a canonical stress-response marker involved in cytoprotection and protein repair) and the pro-apoptotic gene *Bax* were upregulated in a time-dependent manner. *Caspase3* mRNA levels increased significantly at 4 h and remained elevated thereafter, driving apoptotic progression. The anti-apoptotic gene *Bcl-2* exhibited a contrasting trend: its expression decreased in a time-dependent manner starting from 4 h, opposite to the trend of pro-apoptotic genes. In addition, the mRNA expression of *ZC3H10* (involved in the regulation of cellular cold acclimatization) peaked at 4 h.

Based on the above findings, subsequent experiments used a 4 h acute cold stress treatment to further investigate the molecular mechanisms underlying apoptosis induced by acute cold stress and the functions of related genes.

### 3.2. Analysis of Transcriptome Sequencing Results

Transcriptomic analysis identified 1205 differentially expressed genes (DEGs) between the control group (CON) and acute cold stress-treated group (LT), comprising 368 upregulated and 837 downregulated genes (Figure 2A). This provides a critical gene set for investigating cold stress effects on cells. To validate sample reliability, hierarchical clustering analysis of DEGs (Figure 2D) revealed pronounced distinctions between CON and LT groups. Principal component analysis (PCA) further demonstrated tight clustering of CON samples, while LT samples formed a separate cluster distinct from CON (Figure 2B). Validation of 10 randomly selected DEGs by RT-qPCR showed high concordance with sequencing data (Figure 2C), further confirming the accuracy of transcriptome sequencing.

Functionally, Gene Ontology (GO) enrichment analysis (Figure 2E) elucidated cold stress impacts across three domains: In the Biological process dimension, genes were significantly enriched in entries such as “negative regulation of apoptotic process”. In the Cellular component dimension, differentially expressed genes were significantly enriched in nucleoplasm and other subcellular structures. The differential distribution of genes involved in processes such as rRNA processing, intracellular protein transport, and protein stabilization was observed in different subcellular structures. In the Molecular function dimension, genes related to functions such as “ATP binding” and “protein binding” were significantly enriched.

KEGG enrichment analysis (Figure 2F) showed that “Apoptosis” and “Apoptosis—multiple species” pathways were significantly enriched, which directly confirmed that the apoptosis pathway was activated in BMECs under cold stress. apoptosis pathway was activated under cold stress, and a large number of differentially expressed genes were involved in it, which was direct evidence of the cellular initiation of apoptosis program. At the same time, PI3K-Akt, MAPK, p53, and other signaling pathways indirectly related to apoptosis were also activated, which intertwined with each other to constitute the apoptosis regulatory network of BMECs under acute cold stress. Through multi-faceted analysis, functional annotation, and literature research, the key differential gene ENO1 was identified, and the multi-level apoptotic regulatory network of bovine mammary epithelial cells (BMECs) under acute cold stress was clarified.

### 3.3. ENO1 Regulates Cold Stress-Induced Apoptosis in BMECs

To investigate the role of the *ENO1* gene in cold stress-induced apoptosis, this study detected the mRNA and protein levels of *ENO1* using RT-qPCR and Western blot. The results showed that compared with the non-interfered control group, the mRNA and protein expression levels of *ENO1* in both the control group (37 °C) and cold stress-treated group (4 °C) were significantly decreased in BMECs with *ENO1* expression interference (*p* < 0.05, Figure 3A–C,H,J), indicating successful *ENO1* interference.

To clarify the correlation between *ENO1* and apoptosis in BMECs under acute cold stress conditions, BMECs were co-treated with acute cold stress (4 °C) and *ENO1* interference. results showed that after co-treatment, the mRNA expression of the pro-apoptotic genes *Bax* and *Caspase3* was significantly increased (Figure 3D,E), and the mRNA expression of anti-apoptotic gene *Bcl-2* and cold stress-related gene *HSP90* was significantly decreased (*p* < 0.05, Figure 3F,G). The results of protein level detection showed that the expression level of *HSP90* was significantly reduced (Figure 3I); the ratios of Cleaved/procasp3 and Bax/Bcl-2 were significantly elevated (Figure 3K,L).

## 4. Discussion

In this study, a reliable in vitro model of acute cold stress in bovine mammary epithelial cells (BMECs) was established to investigate the molecular mechanism of apoptosis induced by acute cold stress. mRNA expression of *HSP90*, a classical stress biomarker [36,37], was upregulated with the increase in the cold stress treatment time, which indicated that the model was successfully constructed. Based on the successfully constructed model, in order to further investigate the effect of acute cold stress on apoptosis, we further detected apoptosis-related indexes. Given the significant dysregulation of apoptosis-related genes and the elevated apoptosis rate, we identified 4 h as the critical time point for apoptosis initiation, which provides a key reference for exploring the early apoptotic events triggered by cold stress.

Under acute cold stress, excessive accumulation of intracellular reactive oxygen species (ROS) causes oxidative damage and mitochondrial dysfunction [38]. Mitochondria are the main source of intracellular ROS and play a key role in initiating and executing cell apoptosis. When mitochondrial dysfunction occurs, it triggers the mitochondria-dependent apoptotic pathway, leading to the release of cytochrome c, formation of apoptosomes, and subsequent activation of downstream caspase-3, thereby initiating cell apoptosis [39]. Early studies have demonstrated that acute cold stress can induce apoptosis in various mammalian cells [23,24,25,26], and excessive apoptosis of BMECs exerts negative effects on milk production efficiency and mammary gland health [40]. Consistent with these early findings, sequencing results showed that acute cold stress treatment significantly increased the activation of apoptotic pathways. KEGG functional analysis revealed that the “Apoptosis” and “Apoptosis-multiple species” pathways were significantly enriched, which directly confirms that the apoptotic pathways of BMECs are activated under cold stress. A large number of differentially expressed genes are involved in these pathways, providing direct evidence that cells initiate apoptotic programs. Meanwhile, signaling pathways indirectly related to apoptosis, such as PI3K–Akt, MAPK, and p53, are also activated. These pathways interact with each other, collectively forming a regulatory network for BMEC apoptosis under acute cold stress.

Studies by Ma J et al. have indicated that *ENO1* can reduce intracellular ROS production and cell apoptosis by regulating mitochondrial homeostasis [41]. As a multifunctional protein, *ENO1* participates in the regulation of apoptotic pathways through multiple mechanisms, and its action mechanisms exhibit significant differences in different cell types and pathological conditions. A study on pancreatic ductal adenocarcinoma (PDAC) found that ENO1 mediates cell apoptosis by regulating ERK activation [42]. In a study on hypoxic pulmonary hypertension models, it was observed that *ENO1* improves mitochondrial function in endothelial cells and reduces apoptosis through the PI3K-Akt-mTOR pathway [43]. It is thus hypothesized that *ENO1* may play an important role in coping with mitochondrial damage caused by acute cold stress by affecting mitochondrial function or related apoptotic pathways.

Based on previous reports and sequencing results, we identified *ENO1* as a key regulatory factor. After treating BMECs at 4 °C for 4 h, the expression level of *ENO1* was significantly upregulated, indicating that cells have an active compensatory mechanism. GO annotation revealed that ENO1 has both core glycolytic enzyme activity and apoptosis regulatory potential. From the perspective of cellular metabolism, ATP depletion caused by cold stress further exacerbates the apoptotic process. As a key glycolytic enzyme, the upregulation of *ENO1* may enhance the glycolytic process and maintain ATP production, thereby alleviating the energy crisis caused by cold stress. This suggests its important role in linking energy metabolism and cell survival. In addition, previous studies have confirmed that *ENO1* not only promotes cell survival by regulating glycolysis in breast cancer but also may interact with *HSP70* to protect hepatocytes from heat stress [44,45]. Meanwhile, our experimental results confirmed that interfering with *ENO1* expression exacerbated acute cold stress-induced apoptosis of BMECs, and these results clarified the core regulatory role of *ENO1* in the survival of BMECs under acute cold stress conditions.

Although our research is insightful, there are limitations, including the singularity of the cell model, the failure to consider the interaction between cells in mammary tissues in vivo, the lack of involvement in differences in cellular stress responses under different physiological states, and the failure to explore the role of *ENO1* in other types of stress (such as oxidative stress). Subsequent studies need to verify the results in in vivo models (such as cold-stressed lactating *dairy cows*). Meanwhile, the precise molecular interactions (such as binding partners and pathway crosstalk) through which *ENO1* inhibits apoptosis have not been clarified.

In conclusion, we confirmed that acute cold stress induces apoptosis of BMECs and identified *ENO1* as a key metabolic factor against this damage. This work not only deepens the understanding of the cold stress response mechanism in bovine mammary epithelial cells but also pioneeringly proposes *ENO1* as a potential target for improving the cold resistance of *dairy cows*.

## 5. Conclusions

In this study, an in vitro cold stress model of bovine mammary epithelial cells (BMECs) was established (treated at 4 °C), confirming that 4 h is the key time point for apoptosis activation. This is the first study to reveal that α-enolase (ENO1), as a key regulatory factor, plays a central role in resisting apoptosis of BMECs induced by acute cold stress. This discovery provides an important theoretical basis for in-depth understanding of the apoptotic mechanism of BMECs under acute cold stress, clarifies the value of *ENO1* as a potential target in improving cellular cold resistance, and lays a foundation for subsequent research on cold resistance breeding of *dairy cows*. Subsequent animal experiments are required to further verify the regulatory role of *ENO1* in vivo and explore its specific molecular mechanism. However, this study has provided a new direction for the improvement strategy of cold resistance traits in *dairy cow* gene regulation. The findings are expected to offer theoretical support for pastures to formulate more scientific early-warning and intervention plans against cold waves, thereby helping reduce production losses in winter and further facilitating the sustainable development of animal husbandry.

## Figures and Tables

**Figure 1 animals-15-02559-f001:**
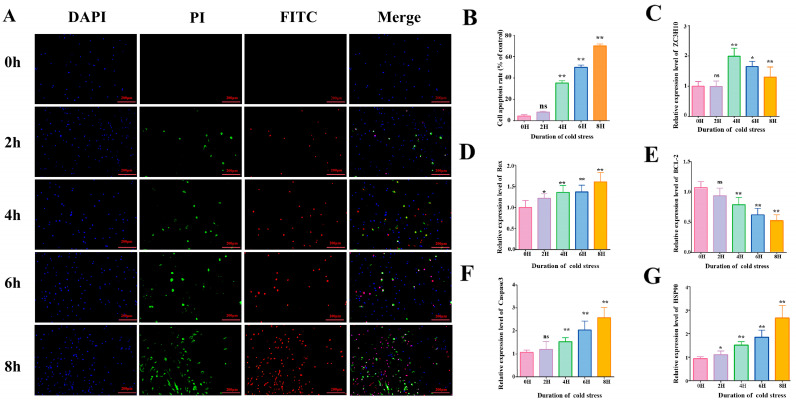
Acute cold stress induces apoptosis. (**A**) Annexin V-FITC fluorescence apoptosis assay of BMECs treated at 4 °C (0 h, 2 h, 4 h, 6 h, 8 h); blue fluorescence represents DAPI-stained cells, green fluorescence represents PI-staining-positive early apoptotic cells, and red fluorescence represents FITC-staining-positive late apoptotic cells. (**B**) Percentage of apoptotic cells. Apoptosis rate = positive apoptotic cells/number of DAPI-stained cells × 100% (*n* = 3) (**C**–**G**) Relative mRNA expression levels of *ZC3H10*, *Bax*, *BCL-2*, *Caspase-3*, and *HSP90* in 4–treated (0 h, 2 h, 4 h, 6 h, 8 h) BMECs (*n* = 6). Data are expressed as mean ± standard error. * *p* < 0.05; ** *p* < 0.01, ns (not significantly different).

**Figure 2 animals-15-02559-f002:**
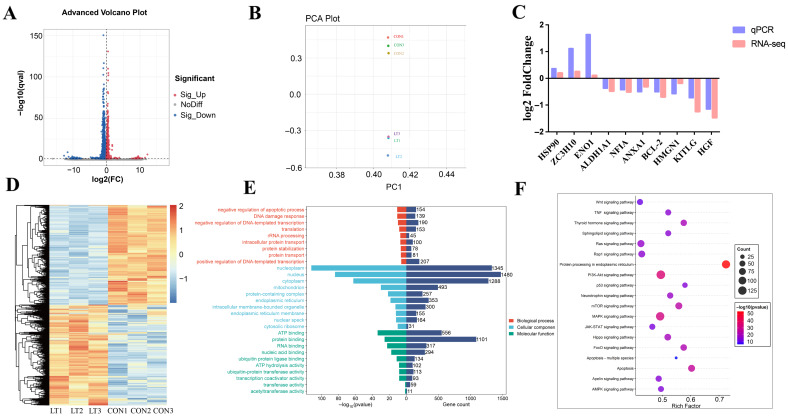
Analysis of differentially expressed genes. (**A**) Volcano plot of differentially expressed genes (DEGs) between the control group (CON) and the 4 h acute cold stress-treated group (LT). (**B**) Principal component analysis (PCA) of the CON group and the 4 h acute cold stress-treated group (LT). (**C**) Validation of 10 DEGs by RT-qPCR. (**D**) Heatmap of DEGs between the CON group and the 4 h acute cold stress-treated group (LT). (**E**) Gene Ontology (GO) enrichment bar plot of DEGs in the CON group versus the 4 h acute cold stress-treated group (LT). (**F**) Kyoto Encyclopedia of Genes and Genomes (KEGG) enrichment bubble plot of DEGs in the CON group versus the 4 h acute cold stress-treated group (LT).

**Figure 3 animals-15-02559-f003:**
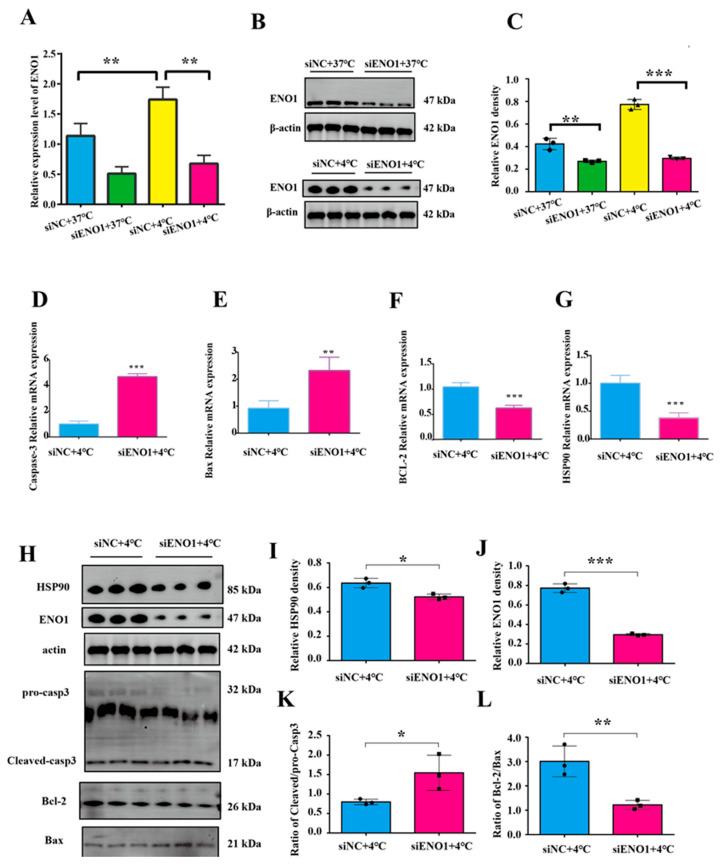
Interference with *ENO1* and acute cold stress co-treatment promotes apoptosis in BMECs. (**A**–**C**) Transfection efficiency of *ENO1* in BMECs. (**D**–**G**) Relative mRNA expression levels of *Caspase3*, *Bax*, *BCL-2*, and *HSP90* in BMECs. (**H**–**L**) Relative protein expression levels of HSP90, ENO1 and apoptosis-related genes in BMECs. The original Western blot (WB) images are available in the Appendix A. (*n* = 3). Data are expressed as mean ± standard error. * *p* < 0.05; ** *p* < 0.01, *** *p* < 0.001, ns (not significantly different).

**Table 1 animals-15-02559-t001:** Primer sequences.

Primer Name	Sequences
β-actin-F	GCCCATCTATGAGGGGTACGC
β-actin-R	CTCCTTGATGTCACGGACGATTTC
BCL-2-F	GCCTTTGTGGAGCTGTATGG
BCL-2-R	CTGTGGGCTTCACTTATGGC
NFIA-F	GAAAGGATCCCACTTCCGGT
NFIA-R	GTCTCCCCACAGCCATCAC
KITLG-F	CCTCTCGTCCACACTCAAGG
KITLG-R	AGTTGTTCCACCATCTCGCT
ALDH1A1-F	GGACCTGTGCAGCAAATCA
ALDH1A1-R	ATAGCAGTTCACCCACACGG
HMGN1-F	CGGGAGGCTTTGATTGTCTTG
HMGN1-R	GCACAACTGACTCCAAACTGC
ANXA1-F	AGTGAGCCCCTATCCTACGTT
ANXA1-R	ACTTCATCCAGGGGCTTTCC
ENO1-F	TGCCACTTATCGGTCATCCTT
ENO1-R	GCGCGTCTTATCATTGTCCC
BAX-F	CCCGAGTTGATCAGGACCAT
BAX-R	GTGGGTGTCCCAAAGTAGGA
CASPASE3-F	TTGAGACAGACAGTGGTGCT
CASPASE3-R	TCTTTGCATTTCGCCAGGAA
ZC3H10-F	GATCTGCTTCTGGTGGGACT
ZC3H10-R	AAAAGGGAACTGACCTGCTGA
HSP90-F	TGCTTGGGAGTCTTCTGCTG
HSP90-R	CACTTCTTTGACCCGCCTCT
HGF-F	TGGCATCAAATGTCAGCCCT
HGF-R	CACCAAGGTCCCCCTTCTTC

Note: Primer name: The identifier for each primer, indicating the gene target and whether it is the forward (F) or reverse (R) strand. Sequence: The nucleotide sequence of each primer.

**Table 2 animals-15-02559-t002:** Sequences of forward and reverse siRNA oligonucleotides targeting *ENO1*.

Name	Primer Sequences
Si-*ENO1*-F	GAGAAGAUCGACAAGCUGA
Si-*ENO1*-R	UCAGCUUGUCGAUCUUCUC

Note: siRNA sequences (5′→3′) targeting Bos taurus ENO1 (NCBI NC__037343). Name: The identifier for each siRNA oligonucleotide, indicating whether it is the forward (F) or reverse (R) strand. Primer Sequences: The nucleotide sequences of the siRNA oligonucleotides.

**Table 3 animals-15-02559-t003:** Antibody dilution concentration.

Antibody	Company	Accession Number	Dilution Ratio
anti-BCL-2	HUABIO (Hangzhou, China)	HA721235	1:5000
anti-BAX	HUABIO (Hangzhou, China)	ET1603-34	1:20,000
anti-HSP90	HUABIO (Hangzhou, China)	HA722689	1:10,000
anti-ENO1	HUABIO (Hangzhou, China)	ET1705-56	1:2000
anti-CYT-C	HUABIO (Hangzhou, China)	ET1701-65	1:2000
anti--β-Actin	GenuIN (Hefei, China)	#2885	1:5000

Note: Antibody: The name of the specific antibody used in the experiments. Company: The supplier of the antibody. Accession Number: The unique identifier provided by the supplier for the specific antibody product. Dilution Ratio: The recommended dilution ratio for the antibody as used in the experiments.

## Data Availability

The original contributions presented in this study are included in the article. Further inquiries can be directed to the corresponding author.

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
