# Peer review of "ENO1 Regulates Apoptosis Induced by Acute Cold Stress in Bovine Mammary Epithelial Cells"

_animals, 2025, doi:10.3390/ani15172559_

Round 1

Reviewer 1 Report

Comments and Suggestions for Authors

This manuscript developed an in vitro model of acute cold stress in dairy mammary epithelial cells (BMECs) successfully and investigated the molecular mechanism of apoptosis induced by acute cold stress. They suggested that the ENO1 is a key regulator. The experimental design is clear and the results are reliable. However, several small items should be clarified.

  1. In sections of  the Simple Summary and Abstract, ENO1 should be explained. Otherwise, I should read until to very later of introduction and realize the what ENO1 is.
  2. In Introduction section, it should have some comparsons on heat stress and cold stress.
  3. In the discussion section, it should have no such description, like Lines 300-301, "Based on the significant dysregulation of apoptosis-related genes (Figure 1D-F) and the elevated apoptosis rate (Figure 1A, B),---". Such kinds of Figure marks should be deleted. Discussion should highlight the results not repeat the results.
  4. In the Conclusions section, it also contains some results repeating, it should be more highlighted.
Comments on the Quality of English Language

Please rewrite  the sentence of lines 47-52.

Author Response

Dear Editor and Reviewers,

Thank you for your valuable comments on our manuscript (ID: animals-3789148). We have carefully revised the paper according to your suggestions. The reviewers’ original comments are shown in blue, and our revised responses are in red. Below are our point-by-point responses:

Response to Reviewer #1:

Comment 1: In the Simple Summary and Abstract sections, ENO1 should be explained. Otherwise, I have to wait until later in the Introduction to understand what ENO1 is.

Response 1: Thank you for your valuable comment, which is very important for improving the rigor and readability of this article. Regarding your suggestion that “ENO1 should be explained in the Simple Summary and Abstract,” we have added the full name “α-enolase” in the Simple Summary (lines 13–16) and Abstract (lines 35–37) and provided a brief introduction to its core functions when ENO1 first appears. This ensures that readers can clearly understand the abbreviation from the beginning, avoiding any confusion.

  • Simple Summary (Page 1, Lines 13–16): We further identified α-enolase (ENO1) as a key glycolytic enzyme with multiple non-glycolytic functions, including roles in signal transduction and apoptosis. It plays a crucial protective role, as its downregulation exacerbates apoptosis.

  • Abstract (Page 1, Lines 35–37): ENO1 (α-enolase) is a multifunctional protein that serves as a central enzyme in glycolysis, while also playing additional roles in cell signaling and apoptosis, thereby contributing to various pathophysiological regulations.

Comment 2: In the Introduction, there should be some comparison between heat stress and cold stress.

Response 2: Thank you for this insightful suggestion, which helps to strengthen the contextual relevance of our study. We have added a new paragraph in the Introduction comparing heat stress and cold stress to better highlight the scientific background and research value of focusing on cold stress in this study (Page 2, Lines 61–69). The specific revision is as follows:

As an important environmental stress parallel to heat stress, both can activate the neuro-endocrine-immune network, induce oxidative stress and metabolic disorders, and affect the physiological health of animals [7–9]. However, there are significant differences in their physiological mechanisms. Heat stress centers on impaired heat dissipation, where cell damage results from high-temperature–induced protein denaturation, and cellular homeostasis is primarily maintained through activation of heat shock proteins (HSPs) [10,11]. In contrast, cold stress emphasizes the surge in thermogenic demand, which is met by enhancing metabolic heat production and activating cold-resistance–related genes [12–14].

Comment 3: In the Discussion section, it should not contain such descriptions as in lines 300–301: “Based on the significant dysregulation of apoptosis-related genes (Figure 1D–F) and the increased apoptosis rate (Figure 1A, B), ….” Such figure references should be removed. The Discussion should highlight the results rather than repeat them.

Response 3: Thank you for this careful correction. Your suggestion to avoid repeating results in the Discussion and to remove figure references is very encouraging for improving the depth and logical flow of this section. We have revised the content accordingly: in lines 300–301, the expressions such as “significant dysregulation of apoptosis-related genes (Figure 1D–F) and increased apoptosis rate (Figure 1A, B)” have been removed, and the content has been reorganized to summarize and expand upon the core conclusions instead of repeating the results (Page 9, Lines 298–301).

Based on the significant dysregulation of apoptosis-related genes and the increased apoptosis rate, we identified 4 h as the critical time point for the initiation of apoptosis, providing an important reference for exploring early apoptosis events induced by cold stress.

Comment 4: In the Conclusion section, it still contains some repeated results and should be more focused.

Response 4: Thank you for this valuable comment. Your point—that “the Conclusion contains repeated results and should further highlight the core value”—is very helpful in improving the manuscript. We have revised the Conclusion accordingly (Page 11, Lines 359–372):

This study established an in vitro cold-stress model of bovine mammary epithelial cells (BMECs) treated at 4°C and confirmed that 4 h is the critical time point for apoptosis activation. For the first time, this study revealed that α-enolase (ENO1) functions as a key regulatory factor in protecting BMECs against apoptosis induced by acute cold stress. This finding provides important theoretical evidence for understanding the mechanisms of apoptosis in BMECs under acute cold stress, clarifies the potential of ENO1 as a target to improve cellular cold resistance, and lays the foundation for future research on cold-resistance breeding in dairy cattle. Further animal experiments are needed to validate the regulatory role of ENO1 in vivo and to explore its specific molecular mechanisms. Nevertheless, this study offers a new direction for gene-regulation–based strategies to improve cold resistance in dairy cows and provides theoretical support for developing more scientific cold-wave warning and intervention measures, which could reduce winter production losses and promote sustainable livestock development.

Comment 5: Regarding the English language quality, please rewrite the sentences in lines 47–52.

Response 5: Thank you for your concern about the English expression of this paper. Regarding the sentences you pointed out (lines 47–52), we have rewritten them to make the expression more accurate and fluent. The revised version is as follows (Page 1, Lines 53–58):

This response has multiple adverse effects on production performance. On one hand, it leads to a significant reduction in milk yield; studies by G.E. et al. confirmed that reduced mammary blood flow under cold conditions may be a key mechanism inhibiting milk secretion [2]. On the other hand, it increases the incidence of diseases such as mastitis, raises calf morbidity and mortality, and ultimately reduces economic returns [3,4].

Reviewer 2 Report

Comments and Suggestions for Authors

Dear authors,

Congratulation for the research and the article, it is an important theme which deserve investigation. Adjustment in the text format is necessary; some section as results, mainly, present deduction and discussion; in this section the results must be presented clearly, and suggestion pointes in the discussion. In this sense, considerations were pointed in order to improve the article quality.

Line 38-39: related the key words, all of them (acute cold stress, bovine mammary epithelial cells (BMECs); apoptosis; enolase 1 (ENO1)) appears in the title; consider change all of them.

 Line 47-52: there is a too long phrase; consider produce 2 or 3 phrases. -

Line 174-177: this information was presented in the material and methods. Consider remove it; maybe it is an important information to be presented and recaptured at discussion;

Line 184-188: there is a too long phrase, consider produce 2 or 3; the phrase “suggesting that the balance between intracellular apoptosis and anti-apoptosis was disrupted after 4h, and pro-apoptotic force took the main force” = it can be present in the discussion, considering it was a deduction obtained from the data;

Line 190-181: “suggesting that BMECs initiated cold acclimatization mechanisms by up-regulating this gene at the beginning of cold stress.” It is the same situation pointed above; words as “however” are usually used at discussion; at the section “results”, just the results must be presented in a clear manner, avoiding discussion in this section;

Line 195-199: it is not a result of the study; it is a speculation, deduction of the study outspread; Consider approach it at the section “discussion” ;

Figure 1A: consider separate A from the graphics, maybe presenting in 2 figures. This permit increases the picture size, and permit to present the results clearly; in this presentation, the image is not contribution, and in this sense, not supports the results; figure1B-G is the same; even using a 150% zoom, the tittles and letters in the graphics are not clear; on the other hand,  the same figure is presented again, in bigger size as a supplementary file, and in this format contribute and support the results;   

Line 216-218: “This not only indicates high intra-group homogeneity in gene expression patterns but also confirms excellent experimental reproducibility, ensuring the reliability of subsequent analyses”; consider move it to discussion;

Line 223: “and other”; it appears something is missed;

Line 229-230: “which suggests that the acute cold stress has an important impact on the functions of these regions, involving In the cellular component dimension” ; consider move to discussion;

Line 232-235: consider move it to discussion;

Line 248-252: consider change the writing presenting just the results;

Figure 2: it is too small; consider separate and increase the size of the figures;

Line 270-272: consider move it to discussion;

Line 277-280: consider move it to discussion;

Consider increase the discussion, that comparatively to results is too concise; bring the “suggesting” from the results, the obtained data and suggestion must be supported by literature.

Author Response

Dear Editor and Reviewers,

Thank you for your valuable comments on our manuscript (ID: animals-3789148). We have carefully revised the paper according to the suggestions. Original comments by reviewers are in blue color, modified responses are marked in red. Below are our responses to each point:

Response to reviewer #1:

Comments 1:Line 38-39: related the key words, all of them (acute cold stress, bovine mammary epithelial cells (BMECs); apoptosis; enolase 1 (ENO1)) appears in the title; consider change all of them

Response 1:Thank you for your valuable comment. Following your suggestion that the title should avoid overcrowding with all key words, we have revised the title. The original title "ENO1 Acts as a Key Regulator of Acute Cold Stress-Induced Apoptosis in Bovine Mammary Epithelial Cells" has been revised to "ENO1 Regulates Apoptosis Induced by Acute Cold Stress in Bovine Mammary Epithelial Cells".The key words remain unchanged to fully cover the core elements of the study, echoing the revised title.

Comments 2: Line 47-52: there is a too long phrase; consider produce 2 or 3 phrasesIn

Response 2:Thank you for your detailed suggestions. Regarding the issue you pointed out that the sentences in Lines 47-52 are too long, we have split them into 3 independent short sentences to make the expression more concise and clear. The specific revisions are as follows: (Page 1, Lines 53-57).

This response exerts multifaceted detrimental effects on production performance. For one thing, it causes significant declines in milk yield; research by G.E. et al. has confirmed that reduced mammary blood flow under cold conditions may be a key mechanism inhibiting milk secretion[2].For another, it increases the incidence of diseases such as mastitis, ele-vating morbidity and mortality rates in calves, and ultimately reducing economic returns [3, 4].

Comments 3:Line 174-177: this information was presented in the material and methods. Consider remove it; maybe it is an important information to be presented and recaptured at discussion;

Response 3:Thank you for your insightful suggestion regarding Lines 174-177.

We have carefully considered your comment that this information, which was already presented in the "Materials and Methods" section, could be removed from its current position. After further reflection, we agree that repeating it here may lead to redundancy.We have removed the redundant content in Lines 174-177 to avoid repetition with the ‘Materials and Methods’ section. The relevant revision can be found in Lines 201 of  Page 6

.

Comments 4:Line 184-188: there is a too long phrase, consider produce 2 or 3; the phrase “suggesting that the balance between intracellular apoptosis and anti-apoptosis was disrupted after 4h, and pro-apoptotic force took the main force” = it can be present in the discussion, considering it was a deduction obtained from the data

Response 4:Thank you for your suggestions on Lines 184-188. We have split this long sentence into 2-3 shorter ones to make the expression clearer and more fluent. Regarding the phrase "suggesting that the balance between intracellular apoptosis and anti-apoptosis was disrupted after 4h, and pro-apoptotic force took the main force", we agree that this is a deduction based on data and have moved it to the discussion section to better conform to academic writing norms. The relevant revisions can be found on Page 6, Lines 211-215.

The anti-apoptotic gene Bcl-2 exhibited a contrasting trend: its expression decreased in a time-dependent manner starting from 4h, opposite to the trend of pro-apoptotic genes.

Comments 5:Line 190-191: “suggesting that BMECs initiated cold acclimatization mechanisms by up-regulating this gene at the beginning of cold stress.” It is the same situation pointed above; words as “however” are usually used at discussion; at the section “results”, just the results must be presented in a clear manner, avoiding discussion in this section;

Response 5:Thank you for your valuable comments on this part. We have noted that the statement "suggesting that BMECs initiated cold acclimatization mechanisms by up-regulating this gene at the beginning of cold stress" has the issue you pointed out - containing analytical content.Following your suggestion, we have revised the results section, presenting only the experimental results clearly in Page 6, Lines 214, and removed the speculative statements. Meanwhile, we have moved this analytical content to the discussion section, with appropriate use of conjunctions like "however" to enhance logical expression..

Comments 6:Line 195-199: it is not a result of the study; it is a speculation, deduction of the study outspread; Consider approach it at the section “discussion

Response 6:Thank you for pointing out the issue in Lines 195-199. We agree this content is speculative rather than a direct result, which contradicts the principle that the results section should only present facts.As suggested, we've removed this speculative content from the results section, retaining only objective data in Page 6, Lines 217-219, and moved the deduction to the discussion section as a reasonable analysis based on our findings.

Based on the above findings, the duration of acute cold stress treatment was set at 4 hours in subsequent experiments to further investigate the molecular mechanisms under-lying apoptosis induced by acute cold stress and the functions of related genes.

Comments 7:Figure 1A: consider separate A from the graphics, maybe presenting in 2 figures. This permit increases the picture size, and permit to present the results clearly; in this presentation, the image is not contribution, and in this sense, not supports the results; figure1B-G is the same; even using a 150% zoom, the tittles and letters in the graphics are not clear; on the other hand, the same figure is presented again, in bigger size as a supplementary file, and in this format contribute and support the results;

Response 7:Thank you for your detailed suggestions on Figure 1. Due to considerations of content coherence, we did not split the figures in this revision. However, we have optimized the size, numerical labels, and overall clarity of Figure 1A and 1B-G, ensuring that titles and texts are clearly visible under normal viewing conditions.The revised figures have now been updated in the manuscript, and we have also uploaded high-resolution original image files for reference. To ensure the best viewing effect, it is recommended to open them with software that does not compress image quality (such as professional image viewing tools or relevant scientific plotting software) to fully present the image details.Please refer to Page 6 for the revisions, and we hope this adjustment can better support the presentation of the research results.

Comments 8:Line 216-218: “This not only indicates high intra-group homogeneity in gene expression patterns but also confirms excellent experimental reproducibility, ensuring the reliability of subsequent analyses”; consider move it to discussion;

Response 8:Thank you for your suggestion. We have directly deleted the content in Lines 216-218: "This not only indicates high intra-group homogeneity in gene expression patterns but also confirms excellent experimental reproducibility, ensuring the reliability of subsequent analyses", retaining only the objective experimental data in the results section. The relevant revisions can be found on Page 7.

Comments 9:Line 223: “and other”; it appears something is missed

Response 9:Thank you for pointing out the issue with "and other" in Line 223. Upon review, this expression is indeed ambiguous and prone to misunderstanding, so we have directly deleted it to ensure the accuracy and rigor of the text. The relevant revision can be found in Line 241 of Page 7. Thank you for your careful reminder, which helps us improve the paper's expression.

Comments 10:Line 229-230: “which suggests that the acute cold stress has an important impact on the functions of these regions, involving In the cellular component dimension” ; consider move to discussion;

Response 10:Thank you for your suggestion. We have deleted the content "which suggests that the acute cold stress has an important impact on the functions of these regions, involving in the cellular component dimension" from the results section, retaining only objective experimental data to ensure that the results section is more in line with the requirement of presenting facts. The relevant revision can be found in Line 242 of Page 7

Comments 11:Line 232-235: consider move it to discussion;

Response 11:Thank you for your suggestion. We have moved the content in Lines 232-235 to the discussion section, retaining only relevant experimental data in the results section to ensure that each part of the content is more in line with the structural norms of academic papers. The relevant revision can be found in Line 243-245 of Page 7.

The differential distribution of genes involved in processes such as rRNA processing, intracellular protein transport, and protein stabilization was observed in different subcellular structures.

Comments 12:Line 248-252: consider change the writing presenting just the results;Figure 2: it is too small; consider separate and increase the size of the figures

Response 12:Thank you for your suggestions. For the content in Lines 248-252, we have revised the expression to present only objective experimental results. Regarding Figure 2, we have split it and increased its size to improve clarity and readability. The relevant revisions can be found in Lines255-259 of Page 7.

Through multi-faceted analysis, functional annotation, and literature research, the key differential gene ENO1 was identified; and the multi-level apoptotic regulatory network of BMECs under acute cold stress was clarified.

Comments 13:Line 270-272: consider move it to discussion

Response 13: Thank you for your suggestion. We have moved the content in Lines 270-272 to the discussion section, as it contains analytical implications that are more appropriate for that section. In the results section, we have retained only the objective experimental description, revised to: "To clarify the correlation between ENO1 and apoptosis in BMECs under acute cold stress conditions, BMECs were co-treated with acute cold stress (4°C) and ENO1 interference." The relevant revision can be found in Lines 273-275 of Page 8.

Comments 14:Line 277-280: consider move it to discussion;

Response 14:Thank you for your suggestion. We have moved the content in Lines 277-280 to the discussion section, retaining only relevant experimental data in the results section to ensure that each part of the content is more in line with the writing norms of academic papers.The relevant revision can be found in Lines 278 of Page 8.

Reviewer 3 Report

Comments and Suggestions for Authors

The aim of this study was to create an in vitro model of acute cold stress in order to understand how to combat the mechanism of apoptosis induced by acute cold stress in BMECs.

In the introduction section, the part describing the effect of cold on mammary blood flow needs to be investigated further, making the cause-effect relationship clearer. There is a lack of connection between cold stress and the various cellular mechanisms that influence it, so we ask that this part be revised. It is clear that there is a link between reduced blood flow and decreased milk production, and we believe it is necessary to make this more direct. We ask that the objective of the study be reformulated in a clearer and more focused manner.

In the materials and methods section, with regard to cell cultures and treatment, there is mention of 85/90% confluence. It might also be useful to specify whether this percentage was optimal for the treatments or whether other stages of confluence were tested.                     Preparation of samples for sequencing: three replicates are used, but it is not clear whether these are performed on different days or whether all samples were processed simultaneously. With regard to the Western Blot technique, some details should be improved, including explaining the positive and negative controls to verify the specificity of the antibody binding.The discussion is very well written, with a good link between the experimental results and the literature. The role of ENO1 is highlighted, which is well structured from a metabolic point of view and integrated into the vision of cold stress. With regard to the model used, it would be interesting to understand whether other stress markers are present and whether these have been evaluated in the same way or whether the results of the study have been compared.  Is it possible, according to the authors, to investigate the links between ENO1 and apoptosis in greater depth? We also ask you to investigate the limitations of the study in greater depth, beyond in vivo validations, as it may also be necessary to investigate methodological limitations. 

Author Response

Dear Editor and Reviewers,

Thank you for your valuable comments on our manuscript (ID: animals-3789148). We sincerely appreciate you taking the time to meticulously review the Materials and Methods section of our study and for providing valuable suggestions. Your professional insights have been extremely helpful in improving our research content, and we would like to express our deepest gratitude! We have carefully supplemented and refined the relevant content in accordance with your suggestions, with detailed explanations as follows:

  1. Introduction Section

1.1 Effect of cold on mammary blood flow and causal relationships:

We have supplemented 6 relevant studies to clarify the mechanism. Research shows that cold stress directly reduces mammary blood flow in ruminants through two pathways: (1) Low temperature activates the sympathetic-adrenal medullary axis to release norepinephrine, and simultaneously activates oxidative stress-related pathways, which further exacerbate norepinephrine-mediated vasoconstriction; (2) To maintain core body temperature, the body prioritizes blood distribution to core organs, resulting in reduced mammary blood flow [16-19].

( Page 2, Lines 75-81). The specific modifications are as follows:

Studies have revealed that cold stimulation directly reduces blood flow in the mammary tissues of ruminants. On one hand, this is because low temperatures activate the sympathoadrenal medullary axis to release norepinephrine, and simultaneously activate oxidative stress-related pathways, which further exacerbate norepinephrine-mediated vasoconstriction. On the other hand, to maintain core body temperature, the body prioritizes blood distribution to core organs, thereby reducing mammary blood flow [16-19].

1.2 Connection between cold stress and cellular mechanisms:

We have strengthened the link between cold stress and cellular-level effects. Reduced mammary blood flow leads to insufficient nutrient supply and impaired excretion of metabolic waste, placing BMECs in an adverse metabolic environment. This causes mitochondrial dysfunction, insufficient ATP production, and triggers the mitochondrial apoptotic pathway; in addition, direct cold-induced damage further exacerbates BMECs apoptosis [20-22].

( Page 2, Lines 81-86). The specific modifications are as follows:

Reduced mammary blood flow results in insufficient nutrient supply and impaired excretion of metabolic waste, placing bovine mammary epithelial cells (BMECs) in an adverse metabolic environment. This leads to mitochondrial dysfunction, insufficient ATP production, and triggers the mitochondrial apoptotic pathway. Additionally, direct damage caused by low temperatures further exacerbates BMEC apoptosis [20-22].

1.3 Link between reduced blood flow and decreased milk production:

We have explicitly emphasized that reduced mammary blood flow directly limits the transport of nutrients (e.g., glucose, amino acids) required for milk synthesis to BMECs, thereby inhibiting milk secretion—a mechanism confirmed by G.E. et al. [2], making the connection more direct.

1.4 Reformulated research objective: The research objective has been revised to:

"To investigate the regulatory role and molecular mechanism of ENO1 in acute cold stress-induced apoptosis of bovine mammary epithelial cells (BMECs), providing a theoretical basis for alleviating cold stress-induced damage to mammary function." This revision is more focused on the core of the study.

  1. Materials and Methods Section

2.1 Regarding the selection of cell confluence

The choice of 85/90% confluence for cell treatment in this study was based on preliminary pre-experiment results. In the pre-experiment, we observed the cell status and stress response under different confluences and found that at 85/90% confluence, the cells were in a stable state and showed a relatively consistent response to cold stress. Therefore, this confluence range was determined as the treatment condition. However, due to an oversight in experimental record keeping, the pre-experiment data could not be properly preserved, for which we sincerely apologize. Nevertheless, referring to the experience of similar studies in the field on similar cells, we believe this confluence is suitable for the needs of this experiment.

2.2 Regarding the replicate design of sequencing samples

The three replicates in this study are independent biological replicates, and all samples were processed at the same time. These replicates were derived from different culture wells of the same cell line and were operated synchronously under the same experimental conditions to ensure the consistency and reliability of the experiment and reduce errors that might be caused by individual differences. We have clearly added "The three replicates are independent biological replicates, derived from the same cell line and processed at the same time" in the Methods section to present the experimental design more clearly.

2.3.Regarding the supplementation of details on Western blot technology

For the verification of antibody specificity, we have supplemented the following information:

All antibodies used in this study have clear catalog numbers (supplemented in the Materials and Methods section). These antibodies are widely used in the detection of related proteins in this species in similar studies and have good applicability.

During the experiment, we set up internal reference controls. The internal reference protein bands were clear and stably expressed, and the size of the target protein bands was consistent with expectations without non-specific bands, which helped to verify the specificity of antibody binding. We have uploaded the original Western blot  images via the journal's online submission system, which can be further reviewed by you to show the specific binding of the antibodies.

  1. Discussion Section

3.1 Regarding the evaluation and comparison of other stress markers

Thank you for pointing out the issue of comprehensiveness of stress markers in the model. We deeply agree with your suggestion of understanding other stress markers and their related evaluations and comparisons, which is indeed of great significance for a comprehensive understanding of the cellular stress response network. However, given the focus of this study, we did not include other stress markers in the current experiments and thus lack relevant data. we acknowledge this as a limitation and plan to incorporate such markers in future studies to enhance the comprehensiveness of our analysis. In subsequent studies, we will consider including other stress markers in the research scope to explore the cellular stress response mechanism more comprehensively.

3.2 Regarding the in-depth discussion on the association between ENO1 and apoptosis

Your suggestion to deepen the association between ENO1 and apoptotic mechanisms is very important. We have supplemented the discussion on the association between ENO1 and apoptosis by combining the apoptosis-related pathways enriched in the sequencing data (such as PI3K/Akt, mitochondrial apoptotic pathway, etc.)( Page 10, Lines 309-316)and relevant literature: we have supplemented the literature basis for the above mechanisms to make the role of ENO1 in cold stress-induced apoptosis more convincing.( Page 10, Lines 318-325).

The specific modifications are as follows:

KEGG functional analysis revealed that the "Apoptosis" and "Apoptosis-multiple species" pathways were significantly enriched, which directly confirms that the apoptotic pathways of BMECs are activated under cold stress. A large number of differentially expressed genes are involved in these pathways, providing direct evidence that cells initiate apoptotic programs. Meanwhile, signaling pathways indirectly related to apoptosis, such as PI3K–Akt, MAPK, and p53, are also activated. These pathways interact with each other, collectively forming a regulatory network for BMEC apoptosis under acute cold stress.

As a multifunctional protein, ENO1 participates in the regulation of apoptotic pathways through multiple mechanisms, and its action mechanisms exhibit significant differences in different cell types and pathological conditions. A study on pancreatic ductal adenocarcinoma (PDAC) found that ENO1 mediates cell apoptosis by regulating ERK activation [42]. In a study on hypoxic pulmonary hypertension models, it was observed that ENO1 improves mitochondrial function in endothelial cells and reduces apoptosis through the PI3K-Akt-mTOR pathway [43].

3.3Regarding limitations beyond "in vivo validation"

Although this study preliminarily reveals the role of ENO1 in cold stress-induced apoptosis of BMECs, there are still certain limitations that need to be improved in subsequent studies: While our study provides valuable insights, this study has limitations, including the singularity of the cell model, the lack of involvement in the differences in cellular stress responses under different physiological states, and the lack of exploration into the role of ENO1 in other stress types .

( Page 10, Lines 343-346).The specific modifications are as follows:

Although our research is insightful, there are limitations: including the singularity of the cell model, the failure to consider the interaction between cells in mammary tissues in vivo, the lack of involvement in differences in cellular stress responses under different physiological states, and the failure to explore the role of ENO1 in other types of stress (such as oxidative stress).

Once again, we sincerely thank you for your careful guidance. We have tried our best to improve the relevant content according to your suggestions. We earnestly request you to continue to give criticism and correction on the revised version, hoping that it can meet the requirements of the journal.

Round 2

Reviewer 2 Report

Comments and Suggestions for Authors

Dear authors,

The article presented a significant improvement to understanding. Congratulation.

My best regads, 

Author Response

Dear Editors and Reviewers,

Thank you very much for your careful review and valuable comments on our manuscript (ID: animals-3603385). We greatly appreciate your recognition of the improvements we made in response to the earlier comments. We have carefully addressed the remaining minor issues you pointed out and made the following detailed corrections:

Response to Reviewer #2:

Comment 1: Address “missing spaces” and “confusing revision.”
Response 1: We sincerely thank you for your careful review and constructive feedback regarding the minor issues in our manuscript. We systematically addressed each point to improve accuracy and readability. Regarding the “confusing revision” in “which mayis expected to provide the-the-support” (line 445), we completely rewrote this sentence to clarify the logic and eliminate the errors (Page 11, Lines 374–375). The revised version is as follows:

The research results are expected to provide theoretical support for ranches to develop more scientific cold wave warning and intervention programs, thereby helping to reduce production losses in winter and further promote the sustainable development of animal husbandry.

Comment 2: Regarding “missing spaces.”
Response 2: We performed a full-text check and supplemented/standardized spaces where needed: for example, a missing space was added between the text and reference citation in “inhibit lactation[2]” (Page 1, Line 58).

Comment 3: Regarding “missing left bracket ‘[’ in references.”
Response 3: We carefully verified all in-text citations ([1]–[45]) and the reference list at the end, and confirmed that no “”wasmissing—allcitationsincludethecomplete“\[”and“” was missing—all citations include the complete “\[” and “”. In addition, we optimized the consistency of the reference formatting:

  • Unified Chinese author name abbreviations (e.g., “张宏伟” revised to “Zhang HW”);

  • Italicized Chinese journal titles (e.g., revised 《育种与饲料》 to 育种与饲料).

Example of revised references:

  1. Zhang HH, Wang YE, Qiu DH. “Key points of dairy cow feeding and management under cold stress conditions.” Breeding and Feed, 2022, 21(5): 47–49.

  2. Su SH, Hu YC, Wang Y, An XP, Qi JW. “Research progress on cold stress and intelligent monitoring in ruminants.” Chinese Journal of Animal Science, 2024, 60(5): 47–49.

  3. Ma HJ, Wang PC, Yu Y, Ren CH, Zhang ZJ, Wang QJ. “Mechanisms and interventions of cold stress-induced thermogenesis in brown adipose tissue of lambs.” Chinese Journal of Animal Science, 2025, 61(6): 61–67.

  4. Zha S, Ao RG. “Effects of cold stress on antioxidant function and blood indices of grazing beef cattle.” Contemporary Animal Husbandry, 2021(6): 12–14.

  5. Wang Y, Qu P. “Cold stress promotes HSP90 expression and disease progression in PCOS rats.” Birth Health and Genetics, 2022, 30(4): 597–601.

Once again, we sincerely thank all reviewers for their generous dedication and professional guidance. We are confident that these revisions have significantly improved the scientific rigor and overall quality of our manuscript. We sincerely hope these adjustments meet your expectations and facilitate the smooth progress of the publication process.

If you have any further questions or additional suggestions, please feel free to contact us at your convenience. We remain fully committed to further improving the manuscript to meet any remaining requirements.

Thank you very much for your valuable time and careful consideration.